# Phosphorylated Poly(vinyl alcohol) Electrospun Mats for Protective Equipment Applications

**DOI:** 10.3390/nano12152685

**Published:** 2022-08-04

**Authors:** Diana Serbezeanu, Tăchiță Vlad-Bubulac, Mihaela Dorina Onofrei, Florica Doroftei, Corneliu Hamciuc, Alina-Mirela Ipate, Alexandru Anisiei, Gabriela Lisa, Ion Anghel, Ioana-Emilia Şofran, Vasilica Popescu

**Affiliations:** 1“Petru Poni” Institute of Macromolecular Chemistry, Aleea Gr. Ghica Voda, 41A, 700487 Iasi, Romania; 2Department of Chemical Engineering, Faculty of Chemical Engineering and Environmental Protection, Gheorghe Asachi Technical University of Iasi, Bd. Mangeron 73, 700050 Iasi, Romania; 3Fire Officers Faculty, Police Academy “Alexandru Ioan Cuza”, Morarilor Str. 3, Sector 2, 022451 Bucharest, Romania; 4Department of Chemical Engineering in Textiles and Leather, Gheorghe Asachi Technical University of Iasi, Bd. Mangeron 73, 700050 Iasi, Romania

**Keywords:** PVA, phenyl dichlorophosphate, thermal stability, flame resistance, MCC test, electrospun mats

## Abstract

The development of intelligent materials for protective equipment applications is still growing, with enormous potential to improve the safety of personnel functioning in specialized professions, such as firefighters. The design and production of such materials by the chemical modification of biodegradable semisynthetic polymers, accompanied by modern manufacturing techniques such as electrospinning, which may increase specific properties of the targeted material, continue to attract the interest of researchers. Phosphorus-modified poly(vinyl alcohol)s have been, thus, synthesized and utilized to prepare environmentally friendly electrospun mats. Poly(vinyl alcohol)s of three different molecular weights and degrees of hydrolysis were phosphorylated by polycondensation reaction in solution in the presence of phenyl dichlorophosphate in order to enhance their flame resistance and thermal stability. The thermal behavior and the flame resistance of the resulting phosphorus-modified poly(vinyl alcohol) products were investigated by thermogravimetric analysis and by cone calorimetry at a micro scale. Based on the as-synthesized phosphorus-modified poly(vinyl alcohol)s, electrospun mats were successfully fabricated by the electrospinning process. Rheology studies were performed to establish the optimal conditions of the electrospinning process, and scanning electron microscopy investigations were undertaken to observe the morphology of the phosphorus-modified poly(vinyl alcohol) electrospun mats.

## 1. Introduction

Over the last decade, considerable effort has been directed towards the development of protective equipment with a high degree of comfort [1,2]. The use of nanomaterials in protective gear is one of the most cutting-edge scientific trends that offers opportunities for a significant improvement in safety and quality of life. The conventional fibers used for protective equipment are: cotton, wool, viscose, and some high-performance fibers such as polyamide (Nomex^®^), polyimide (Kevlar^®^ and Twaron^®^), polybenzimidazole (PBI^®^), and polybenzoxazole (Zylon^®^). Due to their huge specific surface area and micro porous structure, nanofibrous materials are used in advanced fields such as protective textiles, nanocomposites, sensors, optical devices, tissue engineering scaffolds, and drug delivery systems [3,4,5,6,7,8,9,10]. For specific protective equipment, for example, firefighters’ gear, it is necessary to use chemically modified materials such as fire-retarded wool and cotton, fire- and high-temperature-resistant polyesters and/or inherently fire-resistant polymers such as aramid, poly(aramid-imide), polyimide, polybenzimidazole, etc. To date, the protective equipment used in the prevention and firefighting sector is heavy, bulky, rigid, and has low vapor permeability. All of these factors inherently have the potential to exacerbate physiological strain and cause fatalities. Moreover, in a fire scenario, the presence of toxic and corrosive micro- and nanoparticles carried by heated atmosphere represents an additional risk. Thus, materials with decreased air permeability are of interest to preserve human lives.

Polyvinyl alcohol is a non-toxic, water-soluble, biocompatible, and biodegradable polymer with broad applications in many industries such as fibers for protective equipment, films and membranes, materials for drug delivery systems, etc. [11,12,13]. However, in order to be used in key area, it must be chemically and/or physically crosslinked [14,15,16,17]. Furthermore, PVA is quite flammable and burns easily. While halogen-based flame retardants will become more restricted, interest in the development and production of innovative eco-friendly solutions for enhancing the flame resistance of polymeric materials remains open. A type of biodegradable polymers known as polyphosphoesters contains the macromolecular function of a phosphoester in which the pentavalent phosphorus atom makes it possible to introduce bioactive molecules that can influence the physical and/or chemical properties of the base polymer [18]. Polyphosphoesters can be classified as polyphosphonates and polyphosphates. Due to the structural versatility given by the presence of the radical units (alkyl/aryl and alkoxy/aryloxy) connected to the phosphorus atom, polymers with varying physical properties and degradation rates can be synthesized. In order to increase the physical properties, polyphosphoesters are commonly copolymerized with polyether and polyesters. Polyphosphoesters and their composites have been utilized in different fields of biology and industry. [19,20,21,22]

In the first instance, partially phosphorylated PVA attracted special attention in research and industry communities due to its improved non-flammability, ability to form complexes with metals, and possibility to produce anionic polyelectrolyte hydrogels or cation exchange resins [23,24]. Currently, partially phosphorylated PVA is being investigated for potential biomedical applications [25]. So far, less research in the literature has been focused on the preparation of micro- and nanofibrous materials from fire retardant polymers containing phosphorus [26,27,28].

The twofold aim here was to chemically crosslink the PVA and to inherently induce flame-retardancy in PVA via a condensation reaction in solution, which takes place between the OH groups along the macromolecular chains of the PVA and P-Cl functionalities present in the phosphorus compound. Thus, three samples, PVA-OP 1 to 3, were synthesized in this study, starting from PVA and phenyl dichlorophosphate. The structure of PVA-OP (1-3) was characterized by Fourier transform infrared spectroscopy (FTIR). The thermal stability of PVA-OP (1-3) was investigated by a thermogravimetric analysis (TGA) and differential scanning calorimetry (DSC). The flame retardancy of PVA-OP (1-3) was evaluated by microscale combustion calorimetry (MCC) tests, while a scanning electron microscopy (SEM) analysis was used to investigate the char residue morphology. Moreover, a rheological investigation was used to identify the optimal concentration for fiber formation. Finally, the PVA-OP (1-3) nanofibers mats were characterized using an SEM analysis.

## 2. Materials and Methods

### 2.1. Materials

PVA powders with different average molecular weights (M_w_ = 9000–10,000 Da (PVA-OP1); M_w_ = 13,000–23,000 Da (PVA-OP2); and M_w_ = 30,000–70,000 Da (PVA-OP3)) and different degrees of hydrolysis (80%; 87–89%; and 87–90%, respectively) were purchased from Sigma-Aldrich (Sigma-Aldrich Chemie GmbH, Eschenstraße 5, 82024 Taufkirchen, Germany). Phenyl dichlorophosphate was purchased from TCI (purity = 98%, TCI EUROPE N.V, Zwijndrecht, Belgium). Dimethylformamide (DMF) was obtained from Sigma-Aldrich (Sigma-Aldrich Chemie GmbH, Eschenstraße 5, 82024 Taufkirchen, Germany). The other chemicals were procured from both external and internal sources and were used as received.

### 2.2. Methods

For the identification of the chemical structure of PVA-OP (1-3), a LUMOS Microscope Fourier Transform Infrared (FTIR) spectrophotometer (Bruker Optik GmbH, Ettlingen, Germany) equipped with an attenuated total reflection (ATR) device was used at frequencies ranging from 4000 to 500 cm^−1^ at a resolution of 4 cm^−1^.

The thermal stability of PVA-OP (1-3) was investigated using Mettler Toledo TGA-SDTA851^e^ equipment in nitrogen atmosphere under dynamic conditions with a flow rate of 20 mL/min and a heating rate of 10 °C/min in the temperature range of 25–750 °C and with a sample mass between 2.34 mg and 3.89 mg.

A differential scanning calorimetry analysis (DSC) was performed with a Mettler Toledo DSC 1 calorimeter (Mettler Toledo, Langacher 44, 8606 Greifensee, Switzerland) with a heating rate of 10 °C/min below a nitrogen flow with flow rate of 150 mL/min. The samples were placed in crucibles of aluminum with perforated caps to allow the evacuation of volatile products.

The flammability behavior of PVA-OP (1-3) was determined using an FTT Micro Calorimeter (Fire Testing Technology Ltd., Holmsfield Rd, Warrington WA1 2DS, United Kingdom). MCC tests were used to evaluate the flammability of PVA-OP (1-3) in controlled temperature conditions; the temperature in the combustor was 900 °C, and the pyrolyser was heated up to 750° at a heating rate of 1 °C/s. The performed tests complied with “Method A” (ASTM D7309-13).

The rheological behavior in a dynamic regime of PVA-OP (1-3) solutions in distilled water was analyzed using a Bohlin Instrument CS-50 rheometer manufactured by Malvern Instruments (Worcestershire, United Kingdom). The measuring system presents cone–plate geometry with a cone angle of 4° and a 40 mm diameter. Shear viscosities were performed in the range of 0.07–50 1/s shear rates at 25 °C. All rheological tests were obtained with an accuracy of ±5% for the various measurements.

A Verios G4 UC Scanning Electron Microscope (Thermo Scientific, Vlastimila Pecha 1282/12, 627 00 Brno-Černovice, Czech Republic) was used to investigate the surface morphology of the PVA-OP (1-3) char residue and for PVA-OP (1-3) electrospun mats. The PVA-OP (1-3) electrospun mats were coated with 6 nm platinum using a Leica EM ACE200 Sputter coater to provide electrical conductivity and to prevent charge buildup during exposure to the electron beam. SEM investigations were performed in high-vacuum mode using a secondary electron detector (Everhart–Thornley detector, ETD) at an accelerating voltage of 5 kV. The diameters of the electrospun fibers were measured by means of the Image J program. At least 25 electrospun fibers from each sample were taken in consideration to obtain the average diameters. In order to evaluate the elemental analysis of the char residue of PVA-OP (1-3), coupled dispersive X-ray spectroscopy (EDX) was used.

The air permeability of the obtained electrospun mats was measured according to ASTM D737-96 using an ATL-2 (Metrimpex-Hungary).

### 2.3. Preparation of the Polymers PVA-OP (1-3)

Polymers PVA-OP (1-3) were prepared by a nucleophilic substitution of the hydroxyl groups of PVA with the -Cl groups present in phenyl dichlorophosphate, according to a method in the literature [25], using DMF as the reaction medium. Figure 1 shows the general structure of PVA-OP. The amount of monomer and polymer used was calculated so that the molar ratio of phosphonic dichloride to PVA was maintained at 1:6 in DMF. Polymers PVA-OP (1-3) were purified by dialysis in distilled water for 5 days. Subsequently, the samples were dried in an oven at 60 °C under vacuum.

### 2.4. Preparation of The Polymer Solution

The PVA-OP (1-3) solutions were prepared by dissolving a certain amount of PVA-OP (1-3) powders in distilled water at room temperature under constant stirring for 5 h in such a manner that the concentrations of the obtained solutions varied between 10 and 30% (*w*/*v*).

### 2.5. Electrospinning Process of PVA-OP (1-3)

The as-obtained PVA-OP (1-3) solutions were electrospun using a Fluidnatek^®^ LE-50 laboratory line from Bioinicia S.L. (Valencia, Spain), equipped with a variable high-voltage 0–30 kV power supply. The solution was electrospun from a 5 mL syringe with a 27G stainless steel needle at a flow rate 50 µL / min. A high voltage around 22 kV was applied when the polymer solution was drowned into fibers and collected on a backer foil sheet attached to a copper grid. The distance between the tip of the needle and the collector was adjusted to 18 cm. The electrospinning process was performed for 2 h at room temperature (25 °C) and at a humidity of 20%.

## 3. Results

### 3.1. FTIR Investigation

The formation of PVA-OP (1-3) networks was confirmed by FTIR spectroscopy. Figure 2 exhibits the FTIR spectra of PVA-OP (1-3). The FTIR spectrum of PVA-OP1 reveals a wide very intense absorption band with a maximum at 3390 cm^−1^, characteristic of the valence vibrations of the hydroxyl O–H bond. This band appeared to be shifted at about 3350 cm^−1^ in the case of the PVA-OP2 and PVA-OP3 samples as a result of their higher molecular weights and degrees of hydrolysis. Absorption bands characteristic of the aliphatic C–H bond in all the samples at 2923/2849 cm^−1^ (asymmetrical and symmetrical valence vibrations) and at 1425 cm^−1^ (deformation vibrations) were also highlighted. The 1728 cm^−1^ absorption band was assigned to the residual acetate groups [29]. In the spectrum of the PVA-OP (1-3) network, the absorption bands located around the value of 1256 cm^−1^ are commonly attributed to the alkyl phosphates (RO)_3_P=O, while the tiny absorption bands appearing at around 1320 cm^−1^ in all the studied samples were attributed to the aryl phosphate (ArO)_3_P=O [30]. Further introspection revealed the presence at approximately 1650 cm^−1^ of bands characteristic of acid phosphates ((RO)_2_(HO)P=O and/or (ArO)_2_(HO)P=O). Signals from 1087 cm^−1^ (asymmetric valence vibrations) and 844 cm^−1^ (symmetric valence vibrations) confirmed the formation of connections in P–O–C linkages [30]. Due to its structural bifunctionality, phenyl dichlorophosphate is expected to function as a crosslinker between PVA chains. Assuming that in the products all kind of alkyl/aryl phosphates and acid phosphates can be formed, as we show graphically in Figure 1, it can be concluded that the chemical crosslinking of the studied PVAs with phenyl dichlorophosphate was successful.

### 3.2. Thermal Stability

The thermal properties of PVA-OP (1-3) were evaluated by a TGA analysis. Table 1 and Figure 3 summarize the data obtained from the thermogravimetric (TG) and differential thermogravimetric (DTG) analyses.

From Table 1, it can be observed that the thermal stability depends on the molecular weight of the PVA. Three distinct stages of decomposition were observed on the DTG thermograms for the samples denoted as PVA-OP1 and PVA-OP2 (Figure 3). The first decomposition peak centered around 56 and 136 °C, respectively. The second stage of decomposition involved temperatures ranging from approximately 292 °C (PVA-OP1) to 303 °C (PVA-OP2), during which the rate of weight loss was very fast, with the loss of mass mainly due to the decomposition of the polymer chains. The third stage for PVA-OP1 and PVA-OP2 was around 451 and 442 °C, respectively. At this stage, the polymer carbonization and the decomposition of the residues took place, a hypothesis supported by the higher weight loss. The lowest value of the char yields was obtained for PVA-OP2. Moreover, in the case of the PVA-OP3, three stages of decomposition were observed. The first was observed at 140 °C, the second at 325 °C, and the third stage at 445 °C. All polymers presented char yields between 4.64 and 10.34% when this parameter was considered at 750 °C. In the case of all polymers in the first stage, it removed the adsorbed moisture from the samples and the traces of solvent [31,32]. The temperature at which the decomposition rate was maximum in the second stage increased with increasing molecular weights of PVA. Similar behavior has been reported by other researchers for PVA with different molecular weights [32].

Glass transition temperatures (Tg) of PVA-OP (1-3) were evaluated by DSC in a temperature range of 25–200 °C at a heating rate of 10 °C/min in nitrogen. As can be seen in Table 1 and Figure 4, the Tg values of PVA-OP (1-3) decreased with increasing polymer average molecular weights of the PVA. By incorporating the phenyl dichlorophosphate into the polymeric matrix, the crosslinking density increased, and therefore the flexibility of the molecular length decreased, especially in the case of low-polymer average-molecular-weight PVA. Thus, the higher Tg value was obtained in the case of PVA-OP1 (Figure 4). In the case of PVA-OP2, the Tg was not detected. The formation of strong hydrogen bonding in PVA-OP2 was probably the main reason for the nonappearance of Tg for this sample. In this case, the amorphous fraction was lower along the macromolecular network, while its semi-crystalline behavior may have obstructed the glass transition during the dynamic scanning calorimetry test.

### 3.3. Morphological Investigation of tbeads only)he Char Residues of PVA-OP (1-3)

Figure 5 shows the SEM images of the PVA-OP2 pyrolysis residues at 342 °C and 475 °C. The residue obtained by heating the sample to 342 °C showed a quasi-stable compact surface. The presence of small intumescent formations could be observed. The appearance of this phenomenon can be attributed to the fact that the speed with which the glassy protective layer is formed from the solid surface is insufficient to prevent the diffusion of volatile products in the pyrolysis zone. The residue obtained at 475 °C was characterized by a relatively smooth and continuous surface that does not allow the diffusion of molecules to the outside or heat to the inner layers, thus preventing advanced decomposition. The presence of the P in the char residue can be observed from the EDX curves for PVA-OP2.

### 3.4. MCC Test

In general, the flammability of materials is characterized by the amount of heat released when the material is exposed to combustion (fire) [33]. The results of MCC analyses for the above samples are summarized in Table 2.

Analyzing the results obtained for the three samples, it can be noticed that from the point of view of HRC and THR, the results are similar. Thus, the difference between the highest HRC and the lowest HRC was only 7.63%. The highest HRC was 265.65 J/(g × K) and belonged to the sample denoted as the PVA-OP1 sample. At the same time, the APV-OP1 sample had the highest char yield of 12.24% but also the lowest THR of 18.24 kJ/g. The lowest HRC was 245.38 J/(g × K) and belonged to the PVA-OP3 sample, which had the lowest char yield of 8.44% and a THR of 19.31 kJ/g, close in value to the PVA-OP2 sample, which was 19.53 kJ/g. In the case of the three samples, the highest HRC did not correspond to the lowest char yield, but vice versa. This may be due to the variable molecular weights that, in the crosslinking form with phenoxy-phosphonic dichloride, cause the samples to behave differently, without a clear hierarchy being established between the three samples.

Although the analyzed samples show PHRR values, in principle, lower than those reported in the literature, from 190 to 350 W/g [34] and even 494 W/g [35], the HRC values are close to those reported by Peng and co-workers [34].

The curves of the all samples are similar, with each sample having two more prominent peaks, similar in size for the PVA-OP1 and PVA-OP3 samples and the first smaller and the second larger for the other sample. From the temperature point of view (Figure 6), it was observed that the stage with the highest heat release, corresponding to the second prominent peak for all three samples, reached the maximum (PHRR) at similar temperatures, approximately 477 °C in the PVA-OP2 sample and approximately 479 °C for the PVA-OP1 and PVA-OP3 samples. Temperature differences were observed at the appearance of the first prominent peaks, at the lowest temperature appearing for the peak of the PVA-OP1 sample, at 317 °C with an HRR of 101.5 W/g, then the peak of the PVA-OP2 sample, with the highest low HRR, about 70 W/g, which occurred at 327 °C, and the peak of the PVA-OP3 sample, 106 W/g, which appeared at a temperature 25 °C higher than that of the PVA-OP1 sample.

The temperature-dependent HRR curves obtained for the studied samples are structurally similar (each sample having, in principle, two main formations) to those of other PVA samples studied in the literature [36,37].

When analyzing the numerical data and the corresponding graphs of the PVA-OP1 test results, no clear hierarchy of samples can be established in terms of fire behavior, and the samples show similar values from the HRC point of view and close values in terms of char yield, THR, PHRR, temperature, and time. Thus, from the HRC point of view, the test with the best fire behavior was the PVA-OP3 test, with an HRC of 245.38 J/(g × K), 7.6% lower than the highest HRC, which was 265.65 J/(g×K) and was recorded in the PVA-OP1 test. From the point of view of char yield, the test with the best fire behavior was PVA-OP1, with a char yield of 12.24%, 3.8% higher than the lowest char yield of 8.44%, which appeared in the PVA-OP1 test.

### 3.5. Rheological Study of PVA-OP (1-3)

In order to understand the influence of PVA-OP (1-3) concentration on the fiber formation process and to select the optimal conditions for the electrospinning process, the viscosity of PVA-OP (1-3) solutions in distillated water was determined in stationary shear conditions (Table 3). By evaluating the rheological parameters, namely, the dynamic viscosity, it was possible to establish and control the optimal concentration of the studied solutions in order to obtain uniform fibers. Thus, the rheological properties were the key factors in electrospinning process, directly affecting the morphology of the electrospun fibers. The viscosity (η) measured at a shear rate of 100 s^−1^ for the sample of PVA-OP3 increased from 0.0289 to 0.7967 Pa × s as the concentration was increased from 3 to 25 wt.% (Table 3). Specific viscosity data were plotted against the PVA-OP3 concentration in order to determine the entanglement concentration, *Ce*, which is the boundary between the semidiluted unentangled regime and the semidiluted entangled regime. The *Ce* in the case of the PVA-OP3 sample was determined to be 12 wt.% from the intercept of the fitted curves in the semidiluted unentangled and the semidiluted entangled regimes (Figure 7). Changes in the slope marked the onset of the semidiluted unentangled and semidiluted entangled regimes. In the semidiluted unentangled regime, the specific viscosity was proportional with c^0.65^, indicating a weak interaction of individual molecules and the absence of significant entanglements, information that is in agreement with the data obtained from SEM. In the semidiluted entangled regime, it was observed that η_sp_ ~ c^2.18^. Moreover, it was observed that after *Ce* fiber formation occurs. At concentrations below 12 wt.%, the increase in the viscosity with increasing concentrations is rather slow, but above 12 wt.%, a slight change occurred, increasing the viscosity rather significantly. From Figure 7, it can be observed that at c ≈ 20 wt.% the formation of uniform fibers without beads occurred, data that are in agreement with the results presented in the paper of McKee and all [38,39]. According to the studies carried out, the authors showed that beaded nanofibers were produced when the solution concentration was greater than or equal to C*e*, and uniform fibers without beads were formed at 2 to 2.5 times C*e* because the chain entanglement became sufficient to form nanofibers.

### 3.6. Electrospinning of PVA-OP (1-3) Solutions

During the electrospinning process of PVA-OP (1-3) the parameters such as the TCD, voltage, and FR were kept constant. The SEM analysis was used to investigate the surface morphology of the PVA-OP(1-3) electrospun mats. Representative SEM images and the average fiber diameters for the electrospun PVA-OP (1-3) membranes are exhibited in Figure 8 and Table 3.

From Figure 8, it can be observed that the average fiber diameters increased with the increases in the PVA average molecular weight (M_w_) and the concentration. Moreover, it can be noticed that at lower M_w_ (in the case of the sample denoted as PVA-OP1), the fibers exhibit a circular cross-section. Uniform fibers were observed at higher M_w_ (in the case of the samples PVA-OP2 and PVA-OP3).

Lower solution concentration (Figure 9 (15% *w*/*v*)) yielded thinner fibers containing beads, while when increasing the solution concentration to 25% for PVA-OP3, uniform fibers were obtained. Therefore, this behavior can be explained by increasing viscoelastic forces that limit the stretching effect of the electrostatic and columbic repulsion forces. For the samples denoted as PVA-OP1, the optimal concentration of 30% *w*/*v* was necessary to obtain the electrospun mats, while for sample PVA-OP2, a concentration of 25% *w*/*v* was necessary to obtain uniform fibers (Figure 9).

The average fiber diameter of the electrospun mats at the optimal condition, investigated using the Image J program, was in the range between 0.111 ± 0.03 and 0.217± 0.045 µm (Table 3).

### 3.7. Air Permeability Measurements

In order to test the air permeability of the electrospun mats, the PVA-OP2 and PVA-OP3 solutions in distilled water were subjected to the electrospinning process for two hours. The air permeability of the PVA-OP2 sample was 0.0973 ± 1.05 × 10^−5^ m^3^/m^2^ × min. Meanwhile, the electrospun mat PVA-OP3 presented an air permeability of 0.0517 ± 2.2 × 10^−5^ m^3^/m^2^ × min. Both samples exhibited comparable average fiber diameters (0.217 ± 0.045 µm for PVA-OP2 and 0.214 ± 0.048 µm for PVA-OP3). Thus, it may be concluded that the electrospun PVA-OP2 is probably more porous. 

## 4. Conclusions

PVAs of different average molecular weights were used to obtain materials based on semisynthetic polymers with improved flame resistance capacity. In this respect, PVA was reacted with phenyl dichlorophosphate in DMF. The structure, morphology, and thermal properties of the phosphorus-containing PVA were investigated. The glass transition temperatures of PVA-OP (1-3) ranged from 58.49 to 67.65 °C, being strictly dependent on the average molecular weight of the starting PVA. The design of PVA-OP (1-3) as polymer materials for smart protective equipment applications relating to heat and flame protection capabilities was also studied. To prepare electrospun mats containing phosphorus, the advanced electrospinning method was applied. The viscosity of PVA-OP (1-3) solutions in distillated water was determined in order to determine the influence of the PVA-OP (1-3) concentration on the fiber formation process, and to select the optimal conditions for the electrospinning process. The PVA-OP (1-3) electrospun mats were characterized by scanning electron microscopy (SEM) and air permeability tests. The results indicated that PVA-OP (1-3) electrospun mats contain varying pore sizes that are dependent on the PVA used in the synthesis.

## Figures and Tables

**Figure 1 nanomaterials-12-02685-f001:**
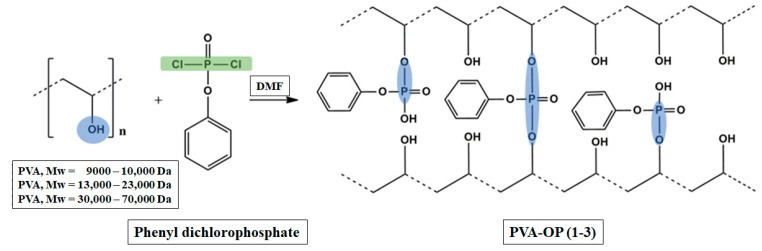
Chemical structure of PVA-OP (1-3).

**Figure 2 nanomaterials-12-02685-f002:**
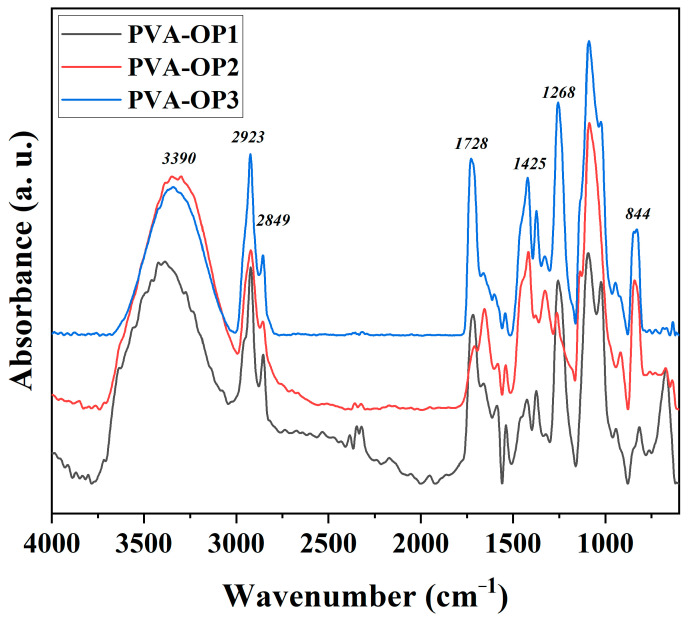
FTIR spectra of the PVA-OP (1-3) samples.

**Figure 3 nanomaterials-12-02685-f003:**
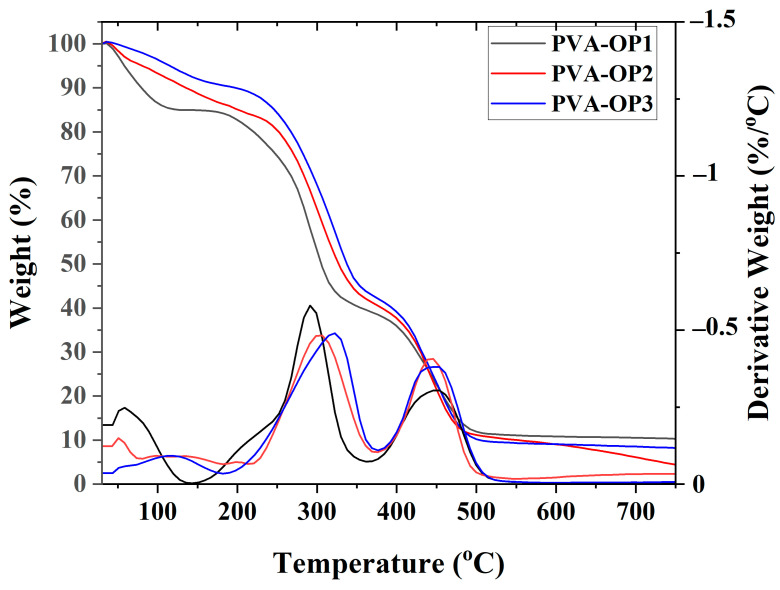
TG and DTG curves for phosphorus-modified poly(vinyl alcohol)s.

**Figure 4 nanomaterials-12-02685-f004:**
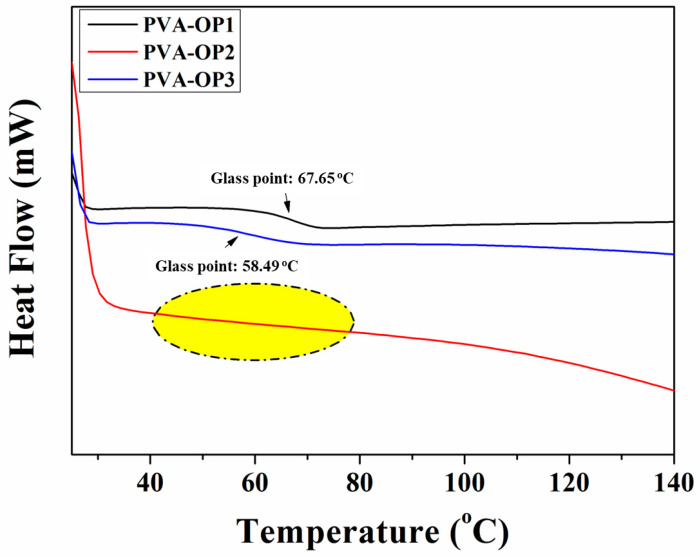
DSC curves for phosphorus-modified poly(vinyl alcohol)s.

**Figure 5 nanomaterials-12-02685-f005:**
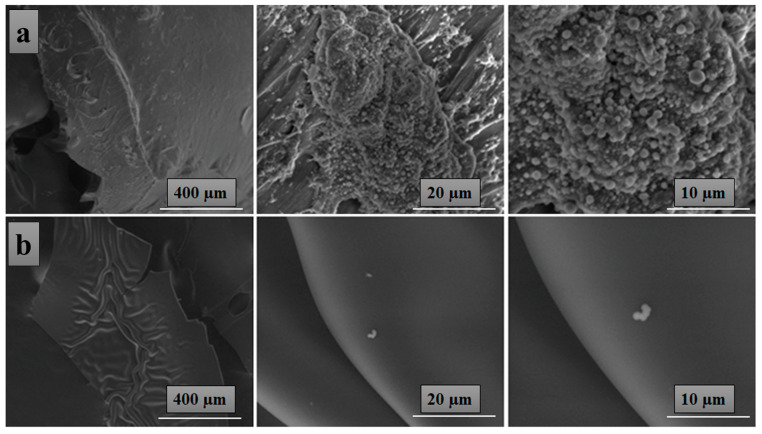
SEM images of PVA-OP2 pyrolysis residues at 342 °C (**a**) and 475 °C (**b**), in the oven of the thermogravimetric analyzer in nitrogen with a heating rate of 10 °C/min.

**Figure 6 nanomaterials-12-02685-f006:**
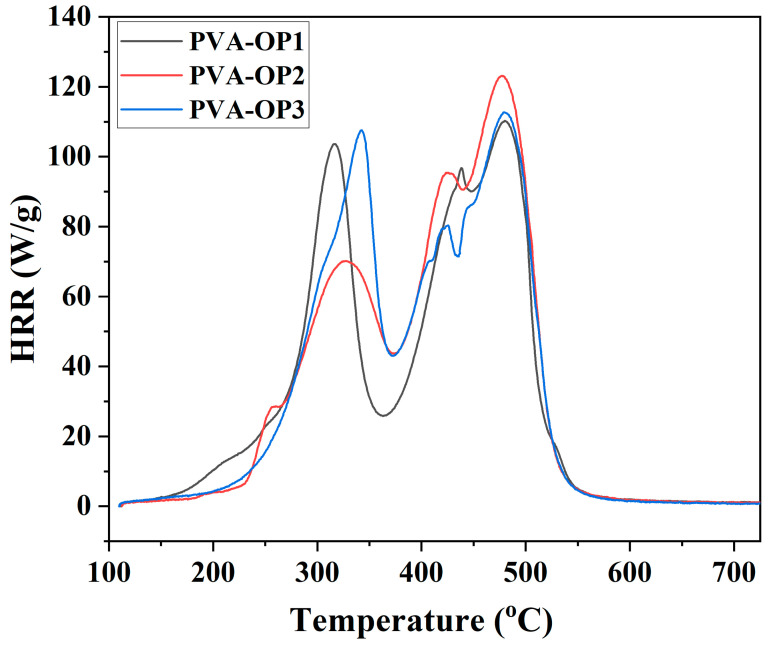
HRR versus temperature for PVA-OP (1-3).

**Figure 7 nanomaterials-12-02685-f007:**
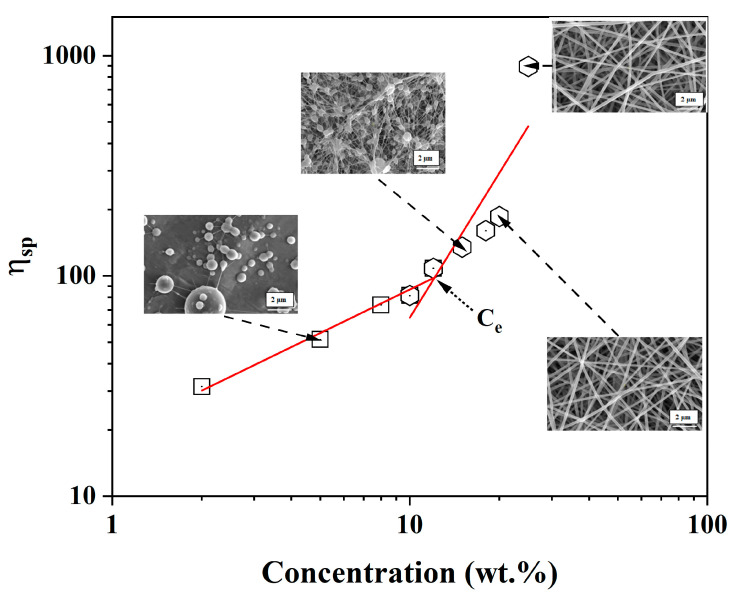
Specific viscosity versus PVA-OP3 concentration in distilled water. SEM images are presented, allowing the visualization of the formation of nanofibers depending on the concentration.

**Figure 8 nanomaterials-12-02685-f008:**
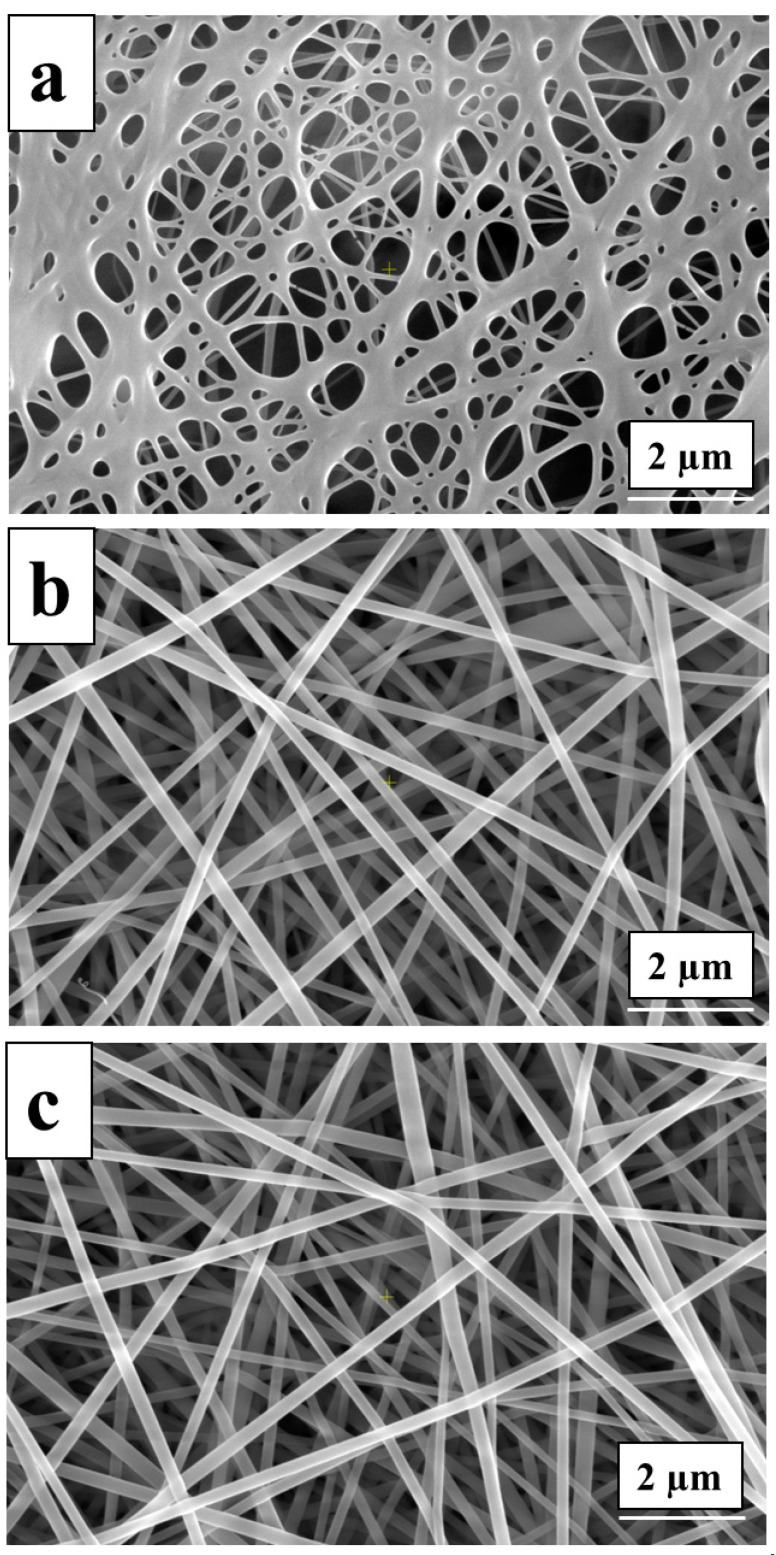
SEM images for the PVA-OP (1-3) electrospun mats at optimal concentration: (**a**) PVA-OP1; (**b**) PVA-OP2; (**c**) PVA-OP3.

**Figure 9 nanomaterials-12-02685-f009:**
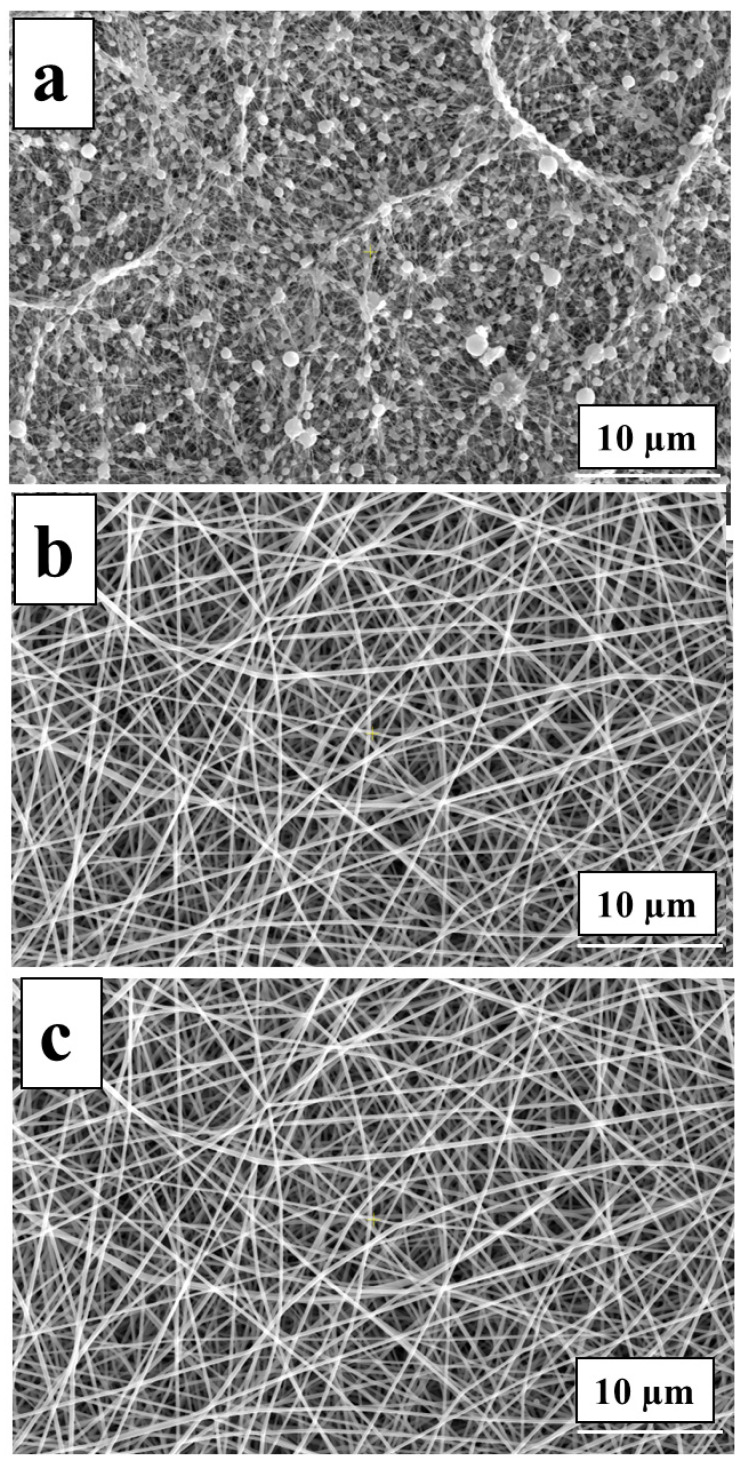
SEM images for PVA-OP3 at different concentrations (15% *w*/*v* (**a**), 20% *w*/*v* (**b**), and 25% *w*/*v* (**c**)).

**Table 1 nanomaterials-12-02685-t001:** Thermal properties of PVA-OP (1-3).

Sample	TGA				DSC
	T_onset_ (°C) ^1^	T_max_ (°C) ^2^	T_endset_ (°C) ^3^	Char Yield (%) ^4^	T_g_ (°C) ^5^
PVA-OP1	46.62 255.65 404.47	56.15 292.41 451.15	101.61 315.20 486.16	10.34	67.65
PVA-OP2	42.10 250.17 416.13	136.36 303.47 442.61	250.17 342.12 475.61	4.64	-
PVA-OP3	52.57 246.92 410.33	140.00 325.51 445.16	150.37 349.30 486.12	8.24	58.49

^1^ initial decomposition temperature; ^2^ the temperature at which the decomposition rate is maximum; ^3^ final decomposition temperature; ^4^ carbonaceous residue yield measured at 750 °C; ^5^ glass transition temperature.

**Table 2 nanomaterials-12-02685-t002:** Data obtained from MCC analysis of poly(vinyl alcohol) samples.

Sample	Weight (mg)	Char Yield (mg)	Char Yield (wt%)	Decomposition Rate (%)	HRC ^1^ (J/(g × K))	THR ^2^ (kJ/g)	PHRR ^3^ (W/g)	T_PHRR_ ^4^ (°C)	Time (s)
PVA-OP1	20.09	2.46	12.24	87.76	265.65	18.24	108.53	479.98	339.50
PVA-OP2	20.07	1.92	9.57	90.43	257.70	19.53	121.83	477.26	336.50
PVA-OP3	20.02	1.69	8.44	91.56	245.38	19.31	111.35	479.20	332.00

^1^ heat release capacity; ^2^ total heat release; ^3^ peak to heat release rate; ^4^ temperature of peak to heat release rate.

**Table 3 nanomaterials-12-02685-t003:** Viscosity, electrospinning conditions (solution concentration, tip–collector distance (TCD), and flow rate (FR) of spinning solution, applied voltage, relative humidity (RH), and ambient temperature), and the average fiber diameters of PVA-OP (1-3).

Sample	Viscosity (Pa × s)	Electrospinning Conditions	Average Fiber Diameters (µm)
PVA-OP1	0.5677	30%, 20 cm, 50 μL/min, 22 kV, 20%, 25 °C	0.111 ± 0.03 (fibers)
PVA-OP2	0.0961	15%, 20 cm, 50 μL/min, 22 kV, 20%, 25 °C	0.056 ± 0.023 (fibers with beads)
	0.6154	25%, 20 cm, 50 μL/min, 22 kV, 20%, 25 °C	0.217 ± 0.045 (uniform fibers)
PVA-OP3	0.0289	2%, 20 cm, 50 μL/min, 22 kV, 20%, 25 °C	- (beads only)
	0.0467	5%, 20 cm, 50 μL/min, 22 kV, 20%, 25 °C	- (beads only)
	0.0666	8%, 20 cm, 50 μL/min, 22 kV, 20%, 25 °C	- (beads with fibers)
	0.0734	10%, 20 cm, 50 μL/min, 22 kV, 20%, 25 °C	- (fibers with beads)
	0.0972	12%, 20 cm, 50 μL/min, 22 kV, 20%, 25 °C	0.031 ± 0.019 (fibers with beads)
	0.1211	15%, 20 cm, 50 μL/min, 22 kV, 20%, 25 °C	0.048 ± 0.020 (fibers with beads)
	0.1437	18%, 20 cm, 50 μL/min, 22 kV, 20%, 25 °C	0.062 ± 0.032 (fibers with beads)
	0.1663	20%, 20 cm, 50 μL/min, 22 kV, 20%, 25 °C	0.304 ± 0.087 (uniform fibers)
	0.7967	25%, 20 cm, 50 μL/min, 22 kV, 20%, 25 °C	0.214 ± 0.048 (uniform fibers)

## Data Availability

Not applicable.

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
