# Peer review of "Phosphorylated Poly(vinyl alcohol) Electrospun Mats for Protective Equipment Applications"

_nanomaterials, 2022, doi:10.3390/nano12152685_

Round 1
Reviewer 1 Report
Abstract should be rewritten. Please, put only facts and dependences here (no samples names1,2,3). If you decide to describe abbreviations in the abstract, you should describe HRC and THR too.
This sentence in the lines 41-24 could mention one more modern application (and corresponding reference) of the electrospun fibers: photocatalyst [DOI: 10.1016/J.NANOEN.2021.106586].
What is the reason of FTIR peaks shift?
Figure 4 should be improved.
Author Response
Reply to the comments of Reviewer 1:
Thank you very much for reviewing our manuscript. We carefully revised the manuscript according to your valuable comments. Our point-by-point replies to the comments of the reviewer and the yellowed changed parts in the revised manuscript were specified below:
Q1: Abstract should be rewritten. Please, put only facts and dependences here (no samples names1,2,3). If you decide to describe abbreviations in the abstract, you should describe HRC and THR too.
A1: The abstract was revised accordingly:
“The development of intelligent materials for protective equipment applications is still growing, having enormous potential to improve the safety of personnel functioning in specialized professions, such as firefighters. The design and production of such materials by chemical modification of biodegradable semi-synthetic polymers accompanied by modern manufacturing techniques such as electrospinning, which may increase specific properties of the targeted material, continues to attract the interest of researchers. Phosphorus-modified poly(vinyl alcohol)s have been, thus, synthesized and utilized to prepare environmentally friendly electrospun mats. Poly(vinyl alcohol)s of three different molecular weights and degrees of hydrolysis have been phosphorylated by polycondensation reaction in solution, in the presence of phenyl dichlorophosphate, in order to enhance their flame resistance and thermal stability. Thermal behavior and the flame resistance of resulted phosphorus-modified poly(vinyl alcohol)s products were investigated by thermogravimetric analysis and by cone calorimetry at micro scale. Based on the as synthesized phosphorus-modified poly(vinyl alcohol)s, electrospun mats have been successfully fabricated by electrospinning process. Rheology studies were performed to establish the optimal conditions of electrospinning process and scanning electron microscopy investigations were undertaken observe morphology of the phosphorus-modified poly(vinyl alcohol)s electrospun mats. ”
Q2: This sentence in the lines 41-24 could mention one more modern application (and corresponding reference) of the electrospun fibers: photocatalyst [DOI: 10.1016/J.NANOEN.2021.106586].
A2: The referred paper has been cited in the revised manuscript.
Q3: What is the reason of FTIR peaks shift?
A3: We have carefully introspected the FTIR spectra of the products and we were able to see a small shift of the band at 3390 cm-1, in the case of PVA-OP1, a band characteristic for valence vibrations of the hydroxyl O–H bonds. This band appeared at about 3350 cm-1, in the case of PVA-OP2 and PVA-OP3, which are samples with a higher degree of hydrolysis. The paragraph about FTIR characterization has been fully reconsidered and rewritten while the Figure 2 has been improved in the revised manuscript:
“The formation of PVA-OP (1-3) networks has been confirmed by FTIR spectroscopy. Figure 2 exhibit the FTIR spectra of PVA-OP (1-3). The FTIR spectrum of PVA-OP1 reveals a wide, very intense absorption band, with a maximum at 3390 cm-1, characteristic of the valence vibrations of the hydroxyl O–H bond. This band appeared shifted at about 3350 cm-1 in the case of the PVA-OP2 and PVA-OP3 samples, as a result of their higher molecular weights and degrees of hydrolysis. Absorption bands characteristic of the aliphatic C–H bond in all the samples at 2923/2849 cm-1 (asymmetrical and symmetrical valence vibrations) and at 1425 cm-1 (deformation vibrations) were also highlighted. The 1728 cm-1 absorption band was assigned to the residual acetate groups [29]. In the spectrum of the PVA-OP (1-3) network, the absorption bands located around the value of 1256 cm-1 are commonly attributed to the alkyl phosphates (RO)3P=O, while the tiny absorption bands appearing at around 1320 cm-1 in all the studied samples, were attributed to the aryl phosphate (ArO)3P=O [30]. Further introspection revealed the presence at approximately 1650 cm-1 of bands characteristic for acid phosphates ((RO)2(HO)P=O and/or (ArO)2(HO)P=O). Signals from 1087 cm-1 (asymmetric valence vibrations) and 844 cm-1 (symmetric valence vibrations) confirmed the formation of connections P–O–C linkages [30]. Due to its structural bifunctionality, phenyl dichlorophosphate is expected to function as crosslinkers between PVA chains. Assuming that in the products there can be formed all kind of alkyl/aryl phosphates and acid phosphates, as we graphically represented in Figure 1, it can be concluded that the chemical crosslinking of studied PVAs with phenyl dichlorophosphate has been successful.
Figure 2. FTIR spectra of the PVA-OP (1-3) samples.”
Q4: Figure 4 should be improved.
A4: An improved Figure 4 has been utilized in the revised manuscript:
Figure 5. SEM images of PVA-OP2 pyrolysis residues at 342 oC (a) and 475 oC (b), respectively, in the oven of the thermogravimetric analyzer, in nitrogen, with a heating rate of 10 oC/min

Reviewer 2 Report
The authors of the manuscript titled: "Phosphorylated Poly(vinyl alcohol) Electrospun Mats for Protective Clothing Applications" propose a method to prepare a thermal resistant material to be used in textile applications. However, there several major issues with the presented research:
1. The prepared materials are stated to be polymers crosslinked by phenyl dichlorophosphate. However, the degree of crosslinking is obviously not enough to produce sufficient polymer network and the material is still water soluble (the authors even use water-based solutions of the polymer for the electrospinning process). It is difficult to imagine a material to be used for production of textiles to be water-soluble. This problem needs to be addressed.
2. The conclusions state that flame resistant material has been achieved. However, the TGA and MCC results show the material starts to decompose at temperatures around 200-250°C. This is comparable to common cotton. The processes taking place at this temperature range in the material need to be studied and the changes of the properties of the material after its exposition to these temperatures need to be described. If the material retains its integrity, the chemical changes of the material after the heating need to be studied. If the material loses its integrity during heating at the 200-250°C range, it can hardly be called flame resistant.
There are also some less severe issues with the manuscript:
1. There are no DSC curves shown in the manuscript and the data obtained from DSC comprise of just glass transition temperatures of 2 of the prepared samples (not even all of the prepared sample types, there is no comment on why the Tg of PVA-OP2 sample could not have been determined). The DSC measurements in this form are superfluous for the presented work, either omit them or expand their presentation/commentary.
2. In Fig. 4 the SEM micrographs are too small. The EDX spectra are then completely unintelligible, they are not marked anyway (one shows the composition of the char and one shows composition of the mat, but it is impossible to tell which is which).
3. In the FTIR spectrum, the absorption band marked at 1268 cm-1 is actually at wavenumber lower than 1250 cm-1. This band is commonly present in pure PVA and cannot be assigned to the presence of the P=O bonds (unless it is a compound band, in which case it shold be commented on). The marking in FTIR spectra by the gray ellipses is also not very clear.
4. Fig. 6 should rather show the dependence of the viscosity on the concentration of the PVA-OP for different samples (PVA-OP 1-3).
5. Fig. 3 is denoted (a), (b) in the caption, but there are no a, b markings in the actual picture.
These are the major issues with the manuscript which need to be adressed if the work is to be further considered for publication.
Author Response
Reply to the comments of Reviewer :
Thank you very much for reviewing our manuscript. We carefully revised the manuscript according to your valuable comments. Our point-by-point replies to the comments of the reviewer and the yellowed changed parts in the revised manuscript were specified below: The authors of the manuscript titled: "Phosphorylated Poly(vinyl alcohol) Electrospun Mats for Protective Clothing Applications" propose a method to prepare a thermal resistant material to be used in textile applications. However, there several major issues with the presented research: Q1: The prepared materials are stated to be polymers crosslinked by phenyl dichlorophosphate. However, the degree of crosslinking is obviously not enough to produce sufficient polymer network and the material is still water soluble (the authors even use water-based solutions of the polymer for the electrospinning process). It is difficult to imagine a material to be used for production of textiles to be water-soluble. This problem needs to be addressed. A1: Indeed, the new polymers are water soluble. We wanted them to show high solubility so that we could electrospun them from green solvent (water). We also synthesized phosphorylated poly(vinyl alcohol) with a higher degree of crosslinking, and the surprise was that these compounds could no longer be processed in the form of fibers, because they presented limited solubility in water. Our main intention here was to produce materials in the form of nanofiber mats appealing for advanced treatment of protective equipment. Of course, the reviewer is wright when questioning the utility of water washable clothing, but that was not the case. It was our error, nevertheless, these kinds of materials are not designed for textile applications, but, instead, they may be used for protective clothing as a key component of the protective equipment, like a filter attached to the protective mask, for example, a filter that may absorb the nanoparticles that are eliminated during combustion, in a fire incident. Therefore, for consistency, the title and “clothing” word in the introduction section have been changed/replaced in the revised manuscript. Q2: The conclusions state that flame resistant material has been achieved. However, the TGA and MCC results show the material starts to decompose at temperatures around 200-250°C. This is comparable to common cotton. The processes taking place at this temperature range in the material need to be studied and the changes of the properties of the material after its exposition to these temperatures need to be described. If the material retains its integrity, the chemical changes of the material after the heating need to be studied. If the material loses its integrity during heating at the 200-250°C range, it can hardly be called flame resistant. A2: Yes, indeed we overrated the flame retardance classification of the materials presented in the current work, we regret for this error. In fact, the products are materials with improved flame retardant capacity when comparing with the neat samples. We edited the first sentence in Conclusions Section accordingly. Besides, we have tested the preservation of the integrity of the electrospun mats after the thermal treatment. As it can be seen from SEM images, provided here for reviewer attention, all the sample are free standing after the treatment at 220oC for 1h. Furthermore, we have repeated the air permeability measurements after thermal treatment of electrospun mats and the initial parameters were preserved. Thus, we may conclude that these types of nanofibers may be used in smart application in construction of key part of protective equipment such as filter masks. Q3. There are no DSC curves shown in the manuscript and the data obtained from DSC comprise of just glass transition temperatures of 2 of the prepared samples (not even all of the prepared sample types, there is no comment on why the Tg of PVA-OP2 sample could not have been determined). The DSC measurements in this form are superfluous for the presented work, either omit them or expand their presentation/commentary. A3: The DSC curves for all the samples were provided in the revised manuscript. We took a look at the polarized light microscope to see differences in the appearance of the nanofibrous mats, at ambient conditions (room temperature), and we were able to observe that the sample PVA-OP2 presented some birefringence, which is a sign of some semi-crystalline nature of the mats, in comparison with the samples PVA-OP1 (a) and PVA-OP3 (c) which showed no birefringence under the observation of the whole microscopic view, suggesting their complete amorphous nature.The discussion regarding DSC measurements has been improved:
“Glass transition temperatures (Tg) of PVA-OP (1-3) were evaluated by DSC on temperature range of 25-200 oC, at a heating rate of 10 oC/min in nitrogen. As it can be seen in Table 1 and Figure 4, the Tg values ​​of PVA-OP (1-3) decreased with increasing the polymer average molecular weight of the PVA. By incorporating the phenyl dichlorophosphate into polymeric matrix, the crosslinking density increased and therefore the flexibility of the molecular length decreased, especially in the case of low polymer average molecular weight PVA. Thus, the higher Tg value was obtained in the case of the PVA-OP1 (Figure 4). In the case of the PVA-OP2 the Tg was not detected. The formation of strong hydrogen bonding in PVA-OP2 probably is the main reason for non appearance of Tg for this sample. In this case, the amorphous fraction is lower along the macromolecular network, while its semi-crystalline behavior may obstruct the glass transition durind dynamic scanning calorimetry test.
Figure 4. DSC curves for phosphorus-modified poly(vinyl alcohol)s.”
Q4: In Fig. 4 the SEM micrographs are too small. The EDX spectra are then completely unintelligible, they are not marked anyway (one shows the composition of the char and one shows composition of the mat, but it is impossible to tell which is which). A4: Figure 4 was changed and a now figure was added in the manuscript. The EDX spectra was removed from the paper:
“
Figure 5. SEM images of PVA-OP2 pyrolysis residues at 342 oC (a) and 475 oC (b), respectively, in the oven of the thermogravimetric analyzer, in nitrogen, with a heating rate of 10 oC/min”
Q5: In the FTIR spectrum, the absorption band marked at 1268 cm-1 is actually at wavenumber lower than 1250 cm-1. This band is commonly present in pure PVA and cannot be assigned to the presence of the P=O bonds (unless it is a compound band, in which case it shold be commented on). The marking in FTIR spectra by the gray ellipses is also not very clear. A5: The gray marking ellipse has been erased to make clear the Figure 2. Actually, we have double checked the FTIR spectra and we can confirm that the referred band is centered at 1255 cm-1 in the case of sample PVA-OP1, at 1261 cm-1 in the case of sample PVA-OP2, and at 1256 cm-1 in the case of sample PVA-OP3. This strong band is commonly attributed to the alkyl phosphates (RO)3P=O, while the tiny absorption bands appearing at around 1320 cm-1 in all the studied samples, is attributed to the aryl phosphate (ArO)3P=O. Further introspection revealed that this assumption is doubled by the presence at approximately 1650 cm-1 of bands characteristic for acid phosphates ((RO)2(HO)P=O and/or (ArO)2(HO)P=O). Assuming that in our products we have all kind of alkyl/aryl phosphates and acid phosphates, as we tried to graphically represent in Figure 1, we consider that our interpretation is now improved in the revised manuscript. The paragraph about FTIR characterization has been fully reconsidered and rewritten while the Figure 2 has been improved in the revised manuscript.Q6: Fig. 6 should rather show the dependence of the viscosity on the concentration of the PVA-OP for different samples (PVA-OP 1-3). A6: Thank you for the recommendation. The figure 7 has been revised showing now the dependence of the specific viscosity on the concentration in the case of PVA-OP3. The other samples behaved the same thus they were easily electrospun using the established parameters.Also, the entire paragraph was added in the manuscript:
“The viscosity (η) measured at shear rate of 100 s-1 for the sample PVA-OP3 increased from 0.0289 to 0.7967 Pa*s as the concentration was increased from 3 to 25 wt% (Table 3). Specific viscosity data were plotted against PVA-OP3 concentration in order to determine the entanglement concentration, Ce, which is the boundary between the semidiluted unentangled regime and the semidiluted entangled regime. The Ce in the case of the sample PVA-OP3 was determined to be 12 wt.% from the intercept of the fitted curves in the semidiluted unentangled and the semidiluted entangled regimes (Figure 7). Changes in the slope marked the onset of the semidiluted unentangled and semidiluted entangled regimes. In the semidiluted unentangled regime, the specific viscosity was proportional with c0.65, indicating weak interaction of individual molecules and absence of significant entanglements, information that are in agreement with the data obtained from SEM. In the semidiluted entangled regime, it can be observed that ηsp ~ c2.18. Also, it can be observed that after Ce, fiber formation occurs. At concentration below 12 wt.% the increase in the viscosity with increasing concentration is rather slow, but above 12 wt.% a slight change is occurred, increasing viscosity rather significatly. From figure 7 it can be observed that at c≈ 20 wt.%, the formation of uniform fibers without beads occures, data that are in agreement with the results presented in the paper of McKee and all. [38,39]. According to the studies carried out, the authors showed that beaded nanofibers were produced when the solution concentration was greater than or equal to Ce, and uniform fibers without beads were formed at 2 to 2.5 times Ce, because the chain entanglement becomes sufficient enough to form nanofibers.”
Figure 7. Specific viscosity versus PVA-OP3 concentration in distilled water.”
Q7: Fig. 3 is denoted (a), (b) in the caption, but there are no a, b markings in the actual picture. A7: The legend of Figure 3 has been rewritten in the revised manuscript: “Figure 3. TG and DTG curves for phosphorus-modified poly(vinyl alcohol)”

Round 2
Reviewer 2 Report
The authors of the manuscript have taken into account and carefully corrected all of the previously stated issues. They showed their due diligence during this process, therefore I recommend the manuscript to be accepted for publication.